# Towards a Simplified and Cost-Effective Diagnostic Algorithm for the Surveillance of Intraductal Papillary Mucinous Neoplasms (IPMNs): Can We Save Contrast for Later?

**DOI:** 10.3390/cancers16050905

**Published:** 2024-02-23

**Authors:** Nicolò Brandi, Matteo Renzulli

**Affiliations:** 1Department of Radiology, Alma Mater Studiorum University of Bologna, 40138 Bologna, Italy; 2Department of Radiology, AUSL Romagna, 48018 Faenza, Italy; 3Department of Radiology, IRCCS Azienda Ospedaliero-Universitaria di Bologna, 40138 Bologna, Italy; dr.matteo.renzulli@gmail.com

**Keywords:** MRCP, imaging, surveillance, intraductal papillary mucinous neoplasm (IPMN), cystic neoplasm, pancreas, contrast medium

## Abstract

**Simple Summary:**

The increased detection of pancreatic cysts in recent years has triggered extensive diagnostic investigations to clarify their potential risk of malignancy, resulting in a large number of patients undergoing numerous imaging follow-up studies for many years. Therefore, there is a growing need for optimization of the current surveillance protocol to provide a practical and reasonable solution in the face of an ever-growing demand. The aim of this paper is to discuss the current available evidence on whether the implementation of unenhanced abbreviated-MRI (A-MRI) protocols for cystic pancreatic lesion surveillance could improve healthcare economics and reduce waiting lists in clinical practice without significantly reducing diagnostic accuracy.

**Abstract:**

The increased detection of pancreatic cysts in recent years has triggered extensive diagnostic investigations to clarify their potential risk of malignancy, resulting in a large number of patients undergoing numerous imaging follow-up studies for many years. Therefore, there is a growing need for optimization of the current surveillance protocol to reduce both healthcare costs and waiting lists, while still maintaining appropriate sensibility and specificity. Imaging is an essential tool for evaluating patients with intraductal papillary mucinous neoplasms (IPMNs) since it can assess several predictors for malignancy and thus guide further management recommendations. Although contrast-enhanced magnetic resonance imaging (MRI) with magnetic resonance cholangiopancreatography (MRCP) has been widely recommended by most international guidelines, recent results support the use of unenhanced abbreviated-MRI (A-MRI) protocols as a surveillance tool in patients with IPMN. In fact, A-MRI has shown high diagnostic performance in malignant detection, with high sensitivity and specificity as well as excellent interobserver agreement. The aim of this paper is, therefore, to discuss the current available evidence on whether the implementation of an abbreviated-MRI (A-MRI) protocol for cystic pancreatic lesion surveillance could improve healthcare economics and reduce waiting lists in clinical practice without significantly reducing diagnostic accuracy.

## 1. Introduction

According to the World Health Organization (WHO), pancreatic cancer represents the 12th most common type of cancer and the 7th most common cause of cancer-related death among the general population, but its incidence is progressively increasing, especially in Western countries, with an average annual growth of 1.1% and estimation of doubling by 2040 [1,2,3,4]. Since most patients with early stages of pancreatic cancer are asymptomatic, this disease tends to be diagnosed only at advanced stages and still yields a very poor prognosis, with only 24% of people surviving 1 year and 9% living for 5 years after diagnosis [5]. Therefore, early diagnosis has now become a recognized healthcare priority [6].

The challenges of early detection include the identification of at-risk populations and the selection of optimal serologic and radiologic modalities for a screening program. As pancreatic cystic lesions may progress to pancreatic ductal adenocarcinoma (PDAC) following the adenoma–carcinoma sequence, with a rough rate of 0.24% per year [7,8,9], and are associated with an increased risk of developing adenocarcinoma elsewhere in the gland (2.2–8.8% at 5 years) [10], patients with these lesions represent the main population to be investigated and monitored [10,11]. Pancreatic cysts can be divided into neoplastic and non-neoplastic types. The latter comprise a group of neoplasms of varying malignant potential and clinical relevance, which are broadly classified according to the World Health Organization (WHO) classification into benign, borderline, and malignant lesions. In particular, intraductal papillary mucinous neoplasms (IPMNs) account for up to 80.9% of pancreatic cysts, followed by serous cystic neoplasms (12.7%) and mucinous cystic neoplasms (2.8%), whereas cystic neuroendocrine tumors and solid pseudopapillary neoplasms are generally regarded as rare lesions (1.5% and 0.7%, respectively) [12,13]. Among pancreatic cystic lesions, mucinous lesions, such as IPMNs and mucinous cystic neoplasms, are those that carry significant malignant potential or might already be malignant at the time they are detected, and thus are those that must be considered for further investigation, surgical resection, and follow-up imaging. IPMNs are divided into three sub-types based on the type of pancreatic duct they involve: branch-duct (BD-IPMN, 70.1%), main-duct (MD-IPMNs, 4.6%), and mixed-type (Mixed-IPMNs, 6.2%) [14,15]. In terms of postoperative pathology, MD-IPMNs and Mixed-IPMNs, regardless of the presence of symptoms, have a high frequency of invasive carcinoma and high-grade dysplasia (61.6%), whereas BD-IPMNs have a much lower incidence of malignancy (31.1%) [10,16].

## 2. Pancreatic Cystic Lesions: An Increasing Burden

The growing use and exponential improvement in the quality of imaging techniques coupled with an aging population has led to a drastic increase in the incidental detection of pancreatic cystic lesions, as testified by the 14-fold increase in the incidence rate of IPMNs between 1985 and 2005 [17,18,19]. Interestingly, the mortality rate for pancreatic adenocarcinoma and IPMN-associated PDAC did not increase over the same period, suggesting a stable number of pancreatic cystic lesions now detected by improved imaging modalities rather than a net increase in incidence [18]. Nonetheless, the true prevalence of these lesions remains unclear and varies significantly between studies due to differences in the timing of the study and the age of the population included [20]. In a meta-analysis encompassing 48,860 patients from 2008 to 2018, the pooled prevalence of incidentally noted pancreatic cysts was 8%, ranging from 2% to 3% in individuals undergoing computed tomography (CT) to 45% in individuals undergoing magnetic resonance imaging (MRI) [21]. Nevertheless, it reached up to 50% in autopsy studies [22]. Of note, a non-negligible proportion of pancreatic cystic lesions, especially those with small diameters, are usually not described in imaging reports in patients without a history of pancreatic disease (69% of cystic lesions with a mean diameter of 6 mm are not reported); thus, their prevalence could be even higher [23].

The increased detection of pancreatic cysts in recent years has triggered extensive diagnostic investigations to clarify their potential risk of malignancy, leading to significant accessibility issues and lengthy waiting lists. This healthcare burden is further amplified by the low estimated malignant progression rate of these lesions, particularly the ones <1 cm, which account for most of the cases [24], resulting in a large number of patients undergoing numerous imaging follow-up studies for many years. In fact, most guidelines state that there are no good long-term data to support the safety of discontinuing surveillance after long-term stability [10]. In addition, these patients also have an increased risk of developing new IPMNs and/or PDAC elsewhere in the whole pancreatic gland (about 1% per year); thus, they must be followed even after the surgical resection of the primary cystic lesion, further increasing the patient cohort requiring repeated imaging follow-up [25,26].

Despite international guidelines being considered cost-effective and clinically appropriate, it is undeniable that the cumulative healthcare expenses will continue to climb as the number of patients with IPMNs participating in follow-up protocols keeps rising [27]. From this perspective, a study published in 2015 estimated that if all patients with pancreatic cysts between 40 and 79 years old underwent MRI surveillance, the median cost would be USD 9.3 billion per year. If we assume that this program was 50% effective in reducing mortality from all PDAC, then this would cost approximately USD 1 million per cancer identified. This is a very conservative estimate because it does not include the cost of endoscopic ultrasound (EUS), further imaging evaluations for staging, and surgery; furthermore, it is not updated with the current epidemiological data [28]. 

Additionally, MRI studies for pancreatic cysts require long examination times and, therefore, take up both human and instrumental valuable resources, reducing their availability for other scopes and potentially engulfing the healthcare systems.

Therefore, in view of the rising detection of pancreatic cystic lesions, there is a growing need for optimization of the current screening and follow-up MRI protocol, in order to reduce both healthcare costs and waiting lists while still maintaining appropriate sensibility and specificity for the detection of malignant transformation. However, as reported by a recent bibliometric analysis, despite a marked increase until 2010, the number of articles regarding IPMN management has recently been somewhat stagnant [29]. The aim of the present paper was to analyze whether the implementation of an abbreviated-MRI (A-MRI) protocol for cystic pancreatic lesion surveillance could improve healthcare economics and reduce waiting lists in clinical practice without significantly reducing diagnostic accuracy.

## 3. Current Management Protocol for Pancreatic Cystic Lesions

The Fukuoka guidelines proposed by the International Association of Pancreatology (IAP) [30] represent one the most followed guidelines in clinical practice since they are universally understood by physicians who treat pancreatic disease as well as by other referring physicians [31,32]. After their first publication in 2006 [33], these guidelines were revised in 2012 [34], 2017 [10], and finally, in 2022 [30] during the 26th meeting of the IAP.

According to the IAP guidelines, several predictors for malignancy must be assessed to guide further management recommendations in patients with IPMNs, based on both clinical and radiological evaluation. Among them, the parameters that can be assessed in surveillance imaging studies are considered the most helpful and significant to establish the actual risk of malignant degeneration. In particular, the presence of an enhancing mural nodule or an enhancing thickened cyst wall have proven to be highly predictive of malignancy since they have been observed in 36–70% and 65% of IPMN patients with invasive disease, respectively [35,36,37,38,39]. In particular, an enhancing mural nodule ≥5 mm is considered a “high-risk stigma” and thus requires surgical resection, whereas an enhancing nodule <5 mm and an enhancing thickened wall are regarded as “worrisome features” and should mandate further evaluation with EUS before intervention. Due to the clinical relevance of these radiological findings, and particularly the importance of discriminating between neoplastic solid components and mucus plugs or debris [40], gadolinium-enhanced MRI with magnetic resonance cholangiopancreatography (MRCP) has been widely recommended by IAP as the procedure of choice for evaluating a pancreatic cyst, based on superior contrast resolution as well as the advantage of avoiding radiation exposure [10,41,42]. Consequently, it is not surprising that the majority of radiologists (70%) support the routine use of intravenous contrast for the characterization and follow-up of incidental pancreatic cystic lesions in clinical practice, as testified by a survey distributed by the Society of Abdominal Radiology Disease Focused Panel [32]. 

## 4. Towards Abbreviated MRI Protocols: Is Contrast Media Really Necessary?

With the current emphasis on cost containment and the increasing rate of incidentally discovered pancreatic cystic lesions, several authors have questioned whether the contrast-enhanced sequences could be eliminated from MRI studies for pancreatic cystic lesions without affecting decision-making. Avoiding the administration of gadolinium through the application of an A-MRI protocol would potentially lead to a number of practical advantages. First, the additional cost of gadolinium would be eliminated [43,44]. Second, as the additional time related to inserting an intravenous catheter into an arm vein, connecting it to a power injector, and performing the contrast-enhanced sequences would no longer be required, the total examination time could be substantially shortened. By doing so, more slots in the daily schedule would become available and could substantially increase the workflow in the MRI suite. Moreover, the decreased time in the magnet would reduce the patient’s potential discomfort in the MRI machine due to claustrophobia and the necessity of lying in a flat position, as well as susceptibility to breathing and movement artifacts secondary to long examinations [45,46,47,48]. Last but not least, concern about the potential risk of nephrogenic systemic fibrosis in patients with borderline renal function would be eliminated, as well as the concern for gadolinium deposition in various body tissues after multiple repeated administrations, thus further decreasing patient anxiety [49,50,51].

In 2009, Macari et al. [52] were the first to initially question whether intravenous contrast was required for MRI follow-up of cystic pancreatic lesions, reporting a retrospective analysis of surveillance MRI studies in 56 patients. The addition of the contrast-enhanced sequences resulted in the same reader interpretation compared to the unenhanced study, yielding a different management recommendation only in five cases (discordance of 4.5%). However, following consensus review, no additional findings that would specifically alter clinical decisions were identified in the contrast-enhanced images and differing recommendations were attributed to expected variations in categorizing lesions. More importantly, no cases concerning malignancy were missed using the A-MRI protocol. Similar results were also reported in the subsequent larger retrospective case review of 301 patients described by Nougaret et al. [43], where follow-up lesion assessment between the unenhanced A-MRI and the classic pancreatic protocol MRI with and without contrast administration was discordant only in 4.6% of cases. Interestingly, the discordant cases represented false-positive rather than false-negative findings and were related to reader misinterpretation of unenhanced T2-w sequences rather than a limitation of the non-enhanced technique. Here, no lesions with suspicious features for malignancy were missed on the A-MRI. In 2017, Pozzi-Mucelli et al. [53] assessed an even shorter surveillance A-MRI protocol in 154 patients consisting essentially of only T1- and T2-weighted sequences. They demonstrated no clinically significant differences in assessment cyst size, main pancreatic duct diameter, or presence of mural nodules between the proposed protocol and the longer comprehensive one, proving how clinical decision-making remained unaffected. In 2020, Kang et al. [54] were the first to report assessing the interobserver agreement and diagnostic accuracy for the presence of “worrisome features” and “high-risk stigmata” recently introduced in the latest Fukuoka revision. In their study, they concluded that there was a substantial inter-reader agreement for evaluating significant imaging features of pancreatic IPMNs using the proposed unenhanced A-MRI protocol, with high sensitivity and negative predictive value. Similarly, in 2022, Johansson et al. [55] reported that their ultra-short MRI protocol correctly allowed all true cystic mural nodules to be detected, with intra- and inter-observer agreement results comparable to the standard contrast-enhanced MRI protocol, but a significant reduction in both examination time and costs. Kierans et al. [56] also confirmed that the addition of gadolinium had no significant impact on the diagnosis of benign versus malignant pancreatic cystic lesions; moreover, they used cytopathology as the reference standard, further validating their results. Finally, Yoo et al. [57] also reported high diagnostic performance with their A-MRI protocol, although the results of subgroup analysis for patients with pathologically confirmed IPMNs revealed lower specificity due to the false-positive risk of mistaking true solid lesions with mucin plugs using unenhanced MRI. Therefore, they concluded that the high sensitivity of the A-MRI protocol regarding mural nodule detection should lead to further evaluation through full-sequence, contrast-enhanced MRI/MRCP or EUS rather than completely replacing them. 

Besides demonstrating comparable efficacy in terms of sensibility and specificity, these relevant studies also confirmed the hypothesized practical advantages of adopting an A-MRI protocol in real clinical scenarios. First, the abbreviated approach resulted in significant cost savings of 61–75% per MRI examination. This would mean an estimated cost reduction by about EUR 50,000 for the potential lifelong surveillance of a 45-year-old patient with a pancreatic cystic lesion that never develops worrisome imaging features [43,53,55]. Second, the A-MRI protocol required only 5–20 min, thus providing an unquestionable reduction in the time of acquisition (from 10 to 27 min according to the standard MRI protocol used) and case reading [43,53,54,55]. Therefore, besides improving the MRI experience for patients, the drastic reduction of scanning time would ensure the execution of roughly 50% more MRI studies, drastically reducing both waiting lists and delays in early diagnosis (Table 1) [58].

Although initial results date back to 2009 and showed promising and practical attractiveness, the most recent revision of the IAP guidelines does not specifically acknowledge the possibility of adopting an A-MRI protocol for the screening and monitoring of pancreatic cystic lesions. Actually, the authors do not even give consideration to this topic or cite the works by Macari et al. [52], Nougaret et al. [43], and Pozzi-Mucelli et al. [53], but rather strongly recommend the use of contrast enhancement in any radiologic examination [30,59]. Specific comments on the potential applicability of an A-MRI scanning protocol for cystic pancreatic lesion surveillance are also lacking in most of the other major clinical guidelines. For example, despite the variations in the specifics (i.e., “solid component” rather than enhancing mural nodules), the American Gastroenterological Association (AGA) Institute Guidelines [25] do not specifically address the issue of whether contrast administration is necessary for an adequate evaluation of pancreatic cysts, and an unspecified “high-quality” MRI with MRCP is proposed as the optimal imaging method in its technical review [8]. Similarly, the American College of Gastroenterology (ACG) 2018 guidelines do not specify whether mural nodules and/or solid components should present enhancement to be considered a high-risk characteristic, nor recommend a specific MRI protocol for pancreatic evaluation [26]. The American College of Radiology (ACR) promotes the execution of a standard pancreatic MRI protocol with contrast-enhanced sequences since it “may help detect enhancement within mural nodules (high-risk stigmata) and the pancreatic phase improves the ability to detect metachronous PDAC elsewhere”. Nonetheless, they at least acknowledge that routinely using contrast material for MRI follow-up is controversial [60]. Finally, despite reporting the results by Macari et al. [52] and Pozzi-Mucelli et al. [53] in evaluating a shortened protocol, the guidelines proposed in 2018 by the European Study Group concluded that “no definite MRI or CT protocol can be recommended for the diagnosis or surveillance of patients with pancreatic cystic lesions because of the wide spread of published data and the lack of dedicated comparative studies” [7]. Intriguingly, only the recent update of the guidelines from the Korean Society of Abdominal Radiology advises towards a shortened MRI protocol for surveillance of pancreatic cystic lesions, especially in patients with impaired renal function since it can provide “sufficient information equivalent to the standard MR protocol”. In particular, based on the previous works by Macari et al. [52] and Nougaret et al. [43], they suggest that a minimum of axial and coronal heavily T2-w and axial T1-w sequences should be performed [61].

The concept of “enhancement” of mural nodules was already mentioned in the second update of the IAP guidelines published in 2012 [34]. Among the cited references regarding the diagnostic work-up for cystic lesions of the pancreas [62,63,64,65,66,67,68,69,70,71], only Ohno et al. [72] have investigated whether the “enhancement” of mural nodules can be related to a higher risk for malignancy but, interestingly, their results were based solely on contrast-enhanced EUS and CT findings. Similarly, among the works cited in the 2017 update [10], only Kang et al. [73] evaluated the enhancement of mural nodules but, again, defined them as protrusions that enhanced after the administration of a contrast agent on CT or present with blood flow on EUS, thus not providing any MRI-related data. Finally, in the latest revision published in 2023 [30], the authors discussed only whether the cut-off value for a mural nodule to be considered a high-risk stigmata should be raised to ≥10 mm. In any case, despite the apparent lack of evidence justifying the need for the administration of contrast medium during MRCP/MRI examinations for IPMNs, it is straightforward that the main problem that could arise from its absence is related to the potential risk for misinterpreting a true neoplastic mural nodule for a false nodule (mucus plugs or debris), and vice versa. Nevertheless, from the data published by Nougaret et al. [43], Pozzi-Mucelli et al. [53], Kang et al. [54], Johansson et al. [55], Kierans et al. [56], and Yoo et al. [57], this risk exists in only sporadic cases and, most importantly, concerns almost exclusively false positives, since true nodules are generally correctly depicted on the T2-w sequences.

According to the IAP guidelines, enhancement evaluation should not be limited to mural nodules but is also a feature of the cystic wall. However, also in this case, by reviewing the works cited in both the 2017 and the 2023 Fukuoka updates in support of wall enhancement evaluation [10,30], none of them were dedicated to the study of this specific feature [63,64,67,73,74,75]. Furthermore, as reported by several studies [54,76,77], cyst wall evaluation shows poor inter-reader agreement even in standard contrast-enhanced MRI protocols due to its subjectivity and measurement error. Therefore, the clinical-pathological significance of an enhancing thickened cyst wall should still be considered controversial and further studies are needed before justifying the need to administer contrast to achieve its characterization.

In addition, the other “high-risk stigmata” and “worrisome features” proposed in the latest update of the Fukuoka guidelines are not affected by the administration or lack of contrast medium during MRI evaluation [30]. First, some of these predictive factors for malignancy, namely the presence of obstructive jaundice, acute pancreatitis, the new onset or acute exacerbation of diabetes within a year, and increased serum level of CA19-9, are an exclusive prerogative of clinicians; therefore, variations in imaging protocols would certainly not affect their assessment. Second, the other radiological parameters that need to be evaluated in patients with IPMNs, such as the diameter of the cyst (≥30 mm), its growth rate (≥2.5 mm/year), the size of the main pancreatic duct (>10 mm or 5–9 mm), and any possible abrupt change in its caliber, can be easily examined in unenhanced imaging. Finally, since the most reliable and reproducible finding suggestive of malignant lymphadenopathy is still size enlargement (generally based on the measurement of the short axis) [78,79], the lack of contrast administration would not significantly influence the diagnostic criteria for suspicious lymph nodes.

In conclusion, therefore, the impact of contrast-enhanced sequences in clinical decision-making for the surveillance of pancreatic cystic neoplasms seems quite limited, for both mural nodules and cystic walls, and thus its administration could potentially be skipped without significant practical implications.

## 5. The Additional Value of Diffusion-Weighted Imaging: Can It Replace the Use of Contrast Agents?

Despite that mural nodularity can be easily seen as a signal blank on T2-w sequences, the use of gadolinium may still have some advantages for malignant risk assessment and follow-up of pancreatic cystic lesions since it may help show enhancement of the nodules through subtraction imaging, thus helping to differentiate them from mucus plugs or debris. Moreover, considering the potential for field defects, the risk of missing a synchronous or metachronous invasive carcinoma elsewhere in the gland in this high-risk population cannot and must not be neglected. Finally, if we consider that, by definition, radiological diagnosis of BD-IPMN can be performed only when a cyst communicating with the main pancreatic duct measures >5 mm in diameter, and that the prevalence of cysts >2 cm is only 0.8% [8], small mural nodules (i.e., those measuring 5–15 mm) might completely occupy the entire volume of the cyst, especially in advanced cases [80,81,82], thus becoming harder to detect solely with heavily T2-w sequences. 

Recently, diffusion-weighted imaging (DWI) has been increasingly used to detect and characterize various tumors, including colon-rectal and hepatic cancers [83]. In light of these results, several authors have attempted to investigate whether DWI can replace or, at least, rationalize the use of contrast agents for the imaging evaluation of patients with pancreatic cystic lesions, thereby improving the sensibility and specificity of MRI investigations. DWI is a functional MRI technique that reflects the Brownian motion of free water in the extracellular, intracellular, and intravascular space. DWI can be assessed in two ways, qualitatively, by visual assessment of signal intensity, and quantitatively, by measurement of the apparent diffusion coefficient (ADC). The ADC value quantifies water proton motion, which in biological tissues is a combination of true water diffusion and capillary perfusion [84]. In general, cancers, including pancreatic cancer, tend to show higher signal intensity on DWI with a lower ADC value than normal tissues because cancer tissues have several factors restricting the water diffusion, such as higher cellularity, tissue disorganization, and extracellular space tortuosity [85]. In particular, in the DWI technique, the histology of different targets varies according to the b-values used, which reflect the strength and timing of the gradients used to generate these particular images [86]. DWI and ADCs with small b-values are generally not very useful in diagnosing malignant tumors since the proportion of diffusion is small and blood perfusion has a greater impact on the final imaging result [87]. Conversely, a high b-value DWI is more sensitive to tissue diffusivity and thus more accurate in detecting malignancy [88]. This is particularly true in the case of IPMNs since their inherent cystic nature leads to a very long T2 relaxation time. Consequently, they might exhibit a strong signal on DWI due to the T2 shine-through effect, which might be mistaken for restricted diffusion. In this context, the use of high b-values (b = 1000 s/mm^2^) is indispensable to correctly predict malignancy [89]. However, there is still no consensus on the true value of ADC measurement appropriate for the evaluation of malignancy of IPMN.

The first attempt to evaluate whether DWI can characterize or predict the malignant potential of cystic pancreatic lesions was performed by Sandrasegaran et al. [90] back in 2011. Although they failed to differentiate malignant from non-malignant lesions, they found that ADC values for benign and low-grade IPMNs were significantly higher than those for high-grade or invasive IPMNs. Afterward, in 2012, Kang et al. [91] evaluated the diagnostic value of DWI for both the malignancy and invasiveness of IPMNs. Interestingly, despite that the addition of DWI significantly improved the diagnostic accuracy for predicting malignancy in only one reader out of three, this additional sequence allowed all observers to decrease the number of false diagnoses made on the basis of the rest of the unenhanced MRI study. Moreover, there was a tendency towards improving the diagnostic accuracy for the prediction of invasive IPMNs with the DWI in two observers. Additionally, Kim et al. [92] demonstrated that diffuse restriction in IPMNs has significantly higher diagnostic accuracy and specificity for predicting malignancy and invasiveness compared to the high-risk stigmata proposed in the 2012 international consensus guidelines [34]. Moreover, it had a tendency to also improve the sensitivity, negative predictive value, and positive predictive value, although it did not show statistical significance. Therefore, based on their encouraging results, the authors concluded that diffusion restriction could be considered as another high-risk stigma of malignancy in patients with IPMNs. Nonetheless, to date, the addition of DWI to the MRI protocol is still not discussed by most international guidelines for the management of pancreatic cystic lesions [25,26,60,61], including the latest Fukuoka update [30]. Only the European Study Group [7] briefly addresses its potential usefulness, affirming that “DWI may be added [...] in order to minimize the risk of missing a concomitant pancreatic cancer”. However, they did not discuss whether DWI can be effectively used for the evaluation of the malignant potential of IPMNs, much less if it can replace the use of contrast agents in this setting.

In the last years, evidence regarding the potential usefulness of DWI for the screening and follow-up of IPMNs has constantly grown. According to a recent meta-analysis [93], DWI can accurately detect the malignant potential of pancreatic IPMNs with an overall pooled sensitivity of 74%, specificity of 94%, and AUC of 0.84. Similarly, another meta-analysis [94] found that the pooled sensitivity, specificity, and AUC of DWI in the differentiation of benign and malignant IPMNs were 72%, 97%, and 0.82, respectively. Additionally, the same study compared the diagnostic performance of the imaging modalities commonly used in clinical practice in distinguishing benign and malignant IPMNs, including CT, MRI/MRCP, PET/CT, EUS, and DWI; in particular, they showed that PET/CT had the highest sensitivity (80%) compared to the other techniques and, more intriguingly, that DWI had the highest pooled specificity (97%). Recently, an attractive study reported comparable diagnostic performance in the identification of mural nodules >5 mm between a dynamic study and DWI, with an accuracy, sensitivity, and specificity compared to the pathological gold standard of 83.02%, 84.62%, and 81.48% for the dynamic study and 79.25%, 75%, and 85.01% for DWI, respectively [95]. Furthermore, as reported by Kang et al. [91], the addition of DWI may potentially improve the agreement between experienced observers and relatively less-experienced observers in unenhanced MR imaging for predicting the malignancy or invasiveness of IPMNs. Nonetheless, studies comparing MRI unenhanced protocols including DWI with the standard contrast-enhanced pancreatic protocol are still lacking. Additionally, despite the encouraging results published by Kierans et al. [56], further studies are needed to elucidate whether DWI can effectively replace contrast-enhanced sequences and thus be included in the A-MRI protocol to reduce the rate of false positives, as well as help in mural nodule characterization.

The potential use of DWI in detecting concomitant invasive carcinoma distant from the IPMN lesion or located away from the pancreatic duct is still under investigation, and evidence is scarce. Kawakami et al. [96] were the only ones who demonstrated that MRCP combined with DWI has a superior diagnostic capability in detecting a concomitant PDAC in patients with IPMNs compared to MRCP alone, reporting both significantly higher sensibility (from 61% to 87–92%) and interobserver agreement (from moderate to almost perfect). Unfortunately, despite being the only one to include DWI among the performed sequences in their A-MRI protocol, the study by Kierans et al. [56] failed to report similar data since no cases of malignancy separate from the pancreatic cystic lesions were evaluated in their cohort. Therefore, whether DWI can effectively alert radiologists to the presence of a separate pancreatic malignancy in patients with IPMNs still requires further analysis.

Despite that DWI could potentially help to achieve a more precise characterization of IPMNs, this technique still yields several technical limitations, and researchers are constantly trying to optimize its acquisition to reach its full potential. First, in the case of tumor-associated obstructive acute pancreatitis, up to 47% of ordinary PDACs might appear indistinguishable from the pancreatic parenchyma distal to the cancer due to the high-viscosity fluid and increased numbers and sizes of inflammatory cells [97]. Second, the slice thickness of DWI is a factor that can significantly affect its diagnostic performance in predicting the malignant potential of IPMNs. Specifically, studies using a thinner slice thickness (5 mm) showed a higher sensitivity compared with those using a thicker slice thickness (7 mm), although without significantly different specificity. Furthermore, studies with quantitative DWI reported a significantly higher specificity compared with those with subjective qualitative DWI, although the pooled sensitivity estimates were not significantly different [84]. Finally, despite that high b-values may improve the detection and classification of solid lesions in IPMNs, the application of increased b-values comes with longer acquisition times and exposition to motion artifacts [98].

## 6. Proposal for a New Algorithm

An appropriate and cost-effective diagnostic algorithm is paramount in determining a population-based approach to pancreatic cystic neoplasm management. In particular, the benefits of early malignant detection must be balanced against the large number of incidental findings. In most cases, in fact, pancreatic cystic lesions have only a remote risk of malignancy and are not associated with any clinical morbidity but still generate high costs for the healthcare system, due to long-term follow-up and expensive diagnostic tests, as well as causing anxiety for patients [34]. Therefore, the primary aim for clinicians should be facilitating the diagnostic process through a pragmatic approach rather than chasing incidental findings with unnecessary expensive investigations, especially in patients harboring relatively common benign lesions with a low risk of malignant transformation. Hence, cost considerations with emphasis on cost-efficacy and utility of long-term surveillance are particularly essential, especially in public healthcare hospitals [99]. 

It Is with this goal in mind that routine use of specific A-MRI protocols could provide a practical and reasonable solution in reducing healthcare costs and improving the MRI workflow in the face of an ever-growing demand; in addition, these optimized protocols would also decrease the anxiety and discomfort of patients due to longer scan times, as well as the related risks that may occur with contrast administration.

Based on the results collected so far, A-MRI protocols could be successfully utilized as a surveillance tool in patients with IPMN since they have demonstrated high diagnostic performance in malignant detection, with high sensitivity and specificity as well as excellent interobserver agreement. In particular, after the first imaging examination, patients with IPMN could be monitored using an unenhanced A-MRI protocol including T1-w images, highly T2-w images (including MRCP), and DWI, which would allow the effective and prompt identification of newly emerging high-risk stigmata and/or worrisome features. The additional inclusion of DWI in the A-MRI protocol could indeed be beneficial to further increase its diagnostic accuracy for early signs of malignancy and, most importantly, to reduce the risk of missing any concomitant PDAC elsewhere in the gland. Furthermore, its addition would not lead to an excessive increase in healthcare costs and the derived relative increase in acquisition time would certainly be inferior to the one derived from the dynamic study. In particular, the A-MRI protocol would require only the administration of one glass of pineapple or blueberry juice which, suppressing the high signal from gastrointestinal tract liquids, would guarantee an adequate image quality of MRCP [100].

Once A-MRI results are available, in doubtful cases and whenever a mural nodule is detected, the patient would be recalled for a personalized, dynamic, contrast-enhanced MRI study only, without the need for repeating previous A-MRI sequences, to differentiate false-positive cases (i.e., mucus plugs or debris) from true malignancies. Going even further, it could even be envisaged a scenario where patients with suspicious findings on A-MRI could directly undergo chest CT and abdominal MRI with gadoxetate disodium (Gd-EOB-DTPA) or gadobenate dimeglumine (Gd-BOPTA). This would ensure a complete contrast-enhanced imaging assessment of both the pancreatic cystic lesion and the liver parenchyma, as well as provide the opportunity to stage the disease simultaneously [101,102,103] (Figure 1). Thereby, the contrast administration would be reserved only for those patients for whom it would actually provide additional useful information. Moreover, time acquisition and cost deriving from the contrast administration during surveillance would be significantly reduced, and a meaningfully higher percentage of patients could be followed up. This approach is consistent with other surveillance strategies, such as the surveillance protocols for hepatocellular carcinoma in cirrhotic patients and breast cancers in women at age risk, where B-mode ultrasound (US) and mammography are successfully used as the first tools for the detection and initial management of tumor occurrence [104,105,106]. 

Despite its encouraging use as a surveillance tool, A-MRI still cannot completely replace the full-sequence, contrast-enhanced protocol on the initial MRI examination based on present scientific and technical knowledge. In fact, the possible imaging findings cannot be anticipated, and an immediate reading of images by radiologists during or right after the acquisition time is unpracticable and potentially fallacious. Nonetheless, it may be worthwhile to investigate whether the proposed A-MRI protocol can also be extended to the screening of patients with suspicious pancreatic cystic lesions and, thus, also be used in their primary characterization. However, additional focused studies that take into account ethical considerations are required.

## 7. Conclusions

The increased detection of pancreatic cysts in recent years has triggered extensive diagnostic investigations to clarify their potential risk of malignancy, resulting in a large number of patients undergoing numerous imaging follow-up studies for many years. Therefore, there is a growing need for optimization of the current surveillance protocol to reduce both healthcare costs and waiting lists, while still maintaining appropriate sensibility and specificity. Imaging is fundamental to evaluating patients with IPMNs since it can assess several predictors for malignancy and thus guide further management recommendations. Despite that gadolinium-enhanced MRI with MRCP has been widely recommended by most international guidelines, recent results support the use of shorter and unenhanced A-MRI protocols as a surveillance tool in patients with IPMN due to the high diagnostic performance in malignant detection, high sensitivity and specificity, as well as excellent interobserver agreement. Reducing both the healthcare costs and acquisition time, the routine use of specific A-MRI protocols could provide a practical and reasonable solution in the face of an ever-growing demand. Nonetheless, further prospective and multicenter studies are needed before these surveillance schemes can be adopted in real-life clinical practice.

## Figures and Tables

**Figure 1 cancers-16-00905-f001:**
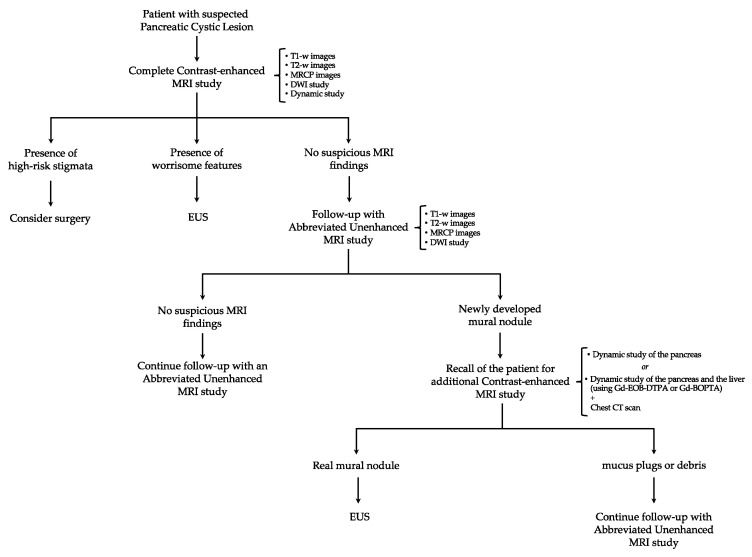
Schematic representation of the proposed new algorithm for the surveillance of patients with pancreatic cystic lesions. MRI: magnetic resonance imaging; MRCP: magnetic resonance cholangiopancreatography; DWI: diffusion-weighted imaging; EUS: endoscopic ultrasound; Gd-EOB-DTPA: gadoxetate disodium; Gd-BOPTA: gadobenate dimeglumine.

**Table 1 cancers-16-00905-t001:** Previous studies that have evaluated the clinical applicability of A-MRI.

	MRI Magnetic Field	A-MRI Protocol Sequences	Time Reduction Compared to Standard Contrast-Enhanced MRI Protocol	Cost Reduction Compared to Standard Contrast-Enhanced MRI Protocol
Macari et al., 2009 [52]	1.5 T	Axial T1-w images STIR/fat-suppressed T2-w images Axial and coronal T2-w images 3D fat-suppressed T1-w images 3D MRCP images	-	-
Nougaret et al., 2014 [43]	1.5 T	Axial T1-w images Axial T2-w images Axial fat-suppressed T2-w images 3D MRCP images	15–20 min. vs. 25–30 min.	-
Pozzi-Mucelli et al., 2017 [53]	1.5 T	Axial and coronal T2-w images Axial 3D fat-suppressed T1-w images	7–8 min. vs. 30–35 min.	EUR 260 vs. 1043 (75%)
Kang et al., 2020 [54]	3 T	Axial 3D fat-suppressed T1-w images Axial and coronal T2-w images 3D MRCP images	5.5 ± 2.1 min. vs. 32.7 ± 8 min.	-
Johansson et al., 2022 [55]	1.5 T	Axial T2-w images 3D MRCP images	7 min. vs. 23 min.	EUR 201 vs. 514 (61%)
Kierans et al., 2022 [56]	1.5 T and 3 T	Axial 3D T1-w images Axial 3D fat-suppressed T1-w images Axial and coronal T2-w images 3D MRCP images DWI (b-values of 0, 50, and 800)	-	-
Yoo et al., 2022 [57]	3 T	Axial and coronal T2-w images 3D MRCP images Axial fat-suppressed T1-w images	5–7 min. vs. 35 min.	-

STIR: Short-Tau Inversion Recovery; MRCP: magnetic resonance cholangiopancreatography.

## Data Availability

The data presented in this study are available upon request from the corresponding author.

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
