# Peer review of "Towards a Simplified and Cost-Effective Diagnostic Algorithm for the Surveillance of Intraductal Papillary Mucinous Neoplasms (IPMNs): Can We Save Contrast for Later?"

_cancers, 2024, doi:10.3390/cancers16050905_

Round 1

Reviewer 1 Report

Comments and Suggestions for Authors

The study proposes a new algorithm for the surveillance of patients with pancreatic cystic lesions, emphasizing cost-effectiveness and utility in clinical practice. It advocates for the routine use of specific A-MRI protocols to improve diagnostic accuracy and reduce healthcare costs. The proposed algorithm involves initial monitoring using unenhanced A-MRI protocols, with dynamic contrast-enhanced MRI reserved for doubtful cases. Despite its promising use as a surveillance tool, the algorithm requires further validation and consideration of practical implementation challenges. Some comments are listed below for the author's consideration:

Section 1:  Specify the sources of the statistics mentioned, such as the studies or databases from which the incidence and mortality rates were derived.

Section 2: The section briefly mentions the potential role of abbreviated-MRI (A-MRI) protocols in reducing healthcare costs and waiting lists without compromising diagnostic accuracy. However, a more in-depth discussion on the evidence supporting A-MRI, its implementation challenges, and its comparative effectiveness with existing protocols would enhance the comprehensiveness of the argument.

- Some statistics mentioned, such as the prevalence of incidentally noted pancreatic cysts and the estimated healthcare costs associated with MRI surveillance, could be further elucidated with details on the studies or methodologies used to derive these figures.

sections 3+4:

-The section presents findings from multiple studies evaluating the efficacy and practical advantages of A-MRI protocols in comparison to traditional contrast-enhanced MRI. This evidence is crucial for assessing the feasibility and impact of adopting A-MRI in clinical practice.

Section 5: While the section effectively discusses the potential of DWI, it could be enhanced by providing more detailed explanations of the technical aspects of DWI, such as the acquisition parameters and interpretation of ADC values. Additionally, including a discussion on the clinical implications of DWI findings and how they may impact patient management could further strengthen the section

Section 6:

-The section effectively communicates the rationale behind the proposed algorithm and its potential benefits in reducing healthcare costs and patient anxiety. The sequence of ideas is logically structured, starting with the problem statement and moving towards the proposed solution.

-The section could benefit from a discussion on the feasibility and challenges of implementing the proposed algorithm in clinical practice. Addressing potential barriers, such as resource constraints or workflow considerations, would add depth to the analysis

Author Response

Dear Reviewer

Please find enclosed the revised version of our manuscript entitled “Towards a Simplified and Cost-Effective Diagnostic Algorithm for the Surveillance of Intraductal Papillary Mucinous Neoplasms (IPMNs): can we save Contrast for later? which we request you to consider for possible publication as a Review Article in Cancers.

Thank you for the opportunity to revise and improve our paper according to your comments. We have modified the main text in accordance with your insightful and significant suggestions and we have replied point by point to all requested revisions.

We hope that now our manuscript reaches a suitable level for a possible publication.

The manuscript, approved by all the Authors, has not been published previously and is not under consideration (in whole or in part) for publication elsewhere.

There is no conflict of interest.

We look forward to hearing from you at your earliest convenience.

Sincerely,

Nicolò Brandi and Matteo Renzulli

Department of Radiology, IRCCS Azienda Ospedaliero-Universitaria di Bologna, Via Albertoni 15, Bologna, Italia.

 The study proposes a new algorithm for the surveillance of patients with pancreatic cystic lesions, emphasizing cost-effectiveness and utility in clinical practice. It advocates for the routine use of specific A-MRI protocols to improve diagnostic accuracy and reduce healthcare costs. The proposed algorithm involves initial monitoring using unenhanced A-MRI protocols, with dynamic contrast-enhanced MRI reserved for doubtful cases. Despite its promising use as a surveillance tool, the algorithm requires further validation and consideration of practical implementation challenges. Some comments are listed below for the author's consideration:

Section 1:  Specify the sources of the statistics mentioned, such as the studies or databases from which the incidence and mortality rates were derived.

RE: Dear Reviewer, thank you very much for your support and meaningful observations. We have checked the statistic and the references mentioned in section 1 and made explicit the source of the data.

Section 2: The section briefly mentions the potential role of abbreviated-MRI (A-MRI) protocols in reducing healthcare costs and waiting lists without compromising diagnostic accuracy. However, a more in-depth discussion on the evidence supporting A-MRI, its implementation challenges, and its comparative effectiveness with existing protocols would enhance the comprehensiveness of the argument.

- Some statistics mentioned, such as the prevalence of incidentally noted pancreatic cysts and the estimated healthcare costs associated with MRI surveillance, could be further elucidated with details on the studies or methodologies used to derive these figures.

RE: Dear Reviewer, thank you very much for your suggestions. However, Reviewer 2 asked us to shorten this section. In section 2, we have tried to summarize the currently available evidence regarding pancreatic cyst management and the estimated healthcare costs associated with MRI surveillance to highlight the purpose of the work. Nonetheless, if the Editor deems it necessary, we can proceed to either shorten or lengthen this section.

sections 3+4:

-The section presents findings from multiple studies evaluating the efficacy and practical advantages of A-MRI protocols in comparison to traditional contrast-enhanced MRI. This evidence is crucial for assessing the feasibility and impact of adopting A-MRI in clinical practice.

Section 5: While the section effectively discusses the potential of DWI, it could be enhanced by providing more detailed explanations of the technical aspects of DWI, such as the acquisition parameters and interpretation of ADC values. Additionally, including a discussion on the clinical implications of DWI findings and how they may impact patient management could further strengthen the section

RE: Dear Reviewer, thank you very much for this suggestion. We have now expanded the part of this section regarding DWI, discussing both the interpretation of ADC values as well as the clinical implication of DWI findings on patient management.

Section 6:

-The section effectively communicates the rationale behind the proposed algorithm and its potential benefits in reducing healthcare costs and patient anxiety. The sequence of ideas is logically structured, starting with the problem statement and moving towards the proposed solution.

-The section could benefit from a discussion on the feasibility and challenges of implementing the proposed algorithm in clinical practice. Addressing potential barriers, such as resource constraints or workflow considerations, would add depth to the analysis

RE: Dear Reviewer, thank you very much for your comment. However, resource constraints are not a barrier to the new proposed algorithm but rather what pushed us to write this work, as we stated in section 4 (lines 150-167). Moreover, workflow considerations have been addressed in section 4, thus discussing them again also in section 6 would result redundant.

Reviewer 2 Report

Comments and Suggestions for Authors

The topic covered by the authors is current and clinically meaningful.

Major remarks

1) The article is very long and some efforts to shorten it should be made, in particular regarding chapter 2.

2) The reference to Fukuoka guidelines is outdated. Please refer to the newly updated guidelines: Ohtsuka T, Fernandez-Del Castillo C, Furukawa T, Hijioka S, Jang JY, Lennon AM, Miyasaka Y, Ohno E, Salvia R, Wolfgang CL, Wood LD. International evidence-based Kyoto guidelines for the management of intraductal papillary mucinous neoplasm of the pancreas. Pancreatology. 2023 Dec 28:S1424-3903(23)01883-5. 

All the text about this citation should be revised accordingly.

3) The authors previously published on the use of pineapple juice for MRI instead of gadolinium. I wonder whether it could be of interest in this setting too.

Author Response

Dear Reviewer

Please find enclosed the revised version of our manuscript entitled “Towards a Simplified and Cost-Effective Diagnostic Algorithm for the Surveillance of Intraductal Papillary Mucinous Neoplasms (IPMNs): can we save Contrast for later? which we request you to consider for possible publication as a Review Article in Cancers.

Thank you for the opportunity to revise and improve our paper according to your comments. We have modified the main text in accordance with your insightful and significant suggestions and we have replied point by point to all requested revisions.

We hope that now our manuscript reaches a suitable level for a possible publication.

The manuscript, approved by all the Authors, has not been published previously and is not under consideration (in whole or in part) for publication elsewhere.

There is no conflict of interest.

We look forward to hearing from you at your earliest convenience.

Sincerely,

Nicolò Brandi and Matteo Renzulli

Department of Radiology, IRCCS Azienda Ospedaliero-Universitaria di Bologna, Via Albertoni 15, Bologna, Italia.

The topic covered by the authors is current and clinically meaningful.

Major remarks

1) The article is very long and some efforts to shorten it should be made, in particular regarding chapter 2.

RE: Dear Reviewer, thank you very much for your suggestions. However, Reviewer 1 asked us to expand this section. In section 2, we have tried to summarize the currently available evidence regarding pancreatic cyst management and the estimated healthcare costs associated with MRI surveillance to highlight the purpose of the work. Nonetheless, if the Editor deems it necessary, we can proceed to either shorten or lengthen this section.

2) The reference to Fukuoka guidelines is outdated. Please refer to the newly updated guidelines: Ohtsuka T, Fernandez-Del Castillo C, Furukawa T, Hijioka S, Jang JY, Lennon AM, Miyasaka Y, Ohno E, Salvia R, Wolfgang CL, Wood LD. International evidence-based Kyoto guidelines for the management of intraductal papillary mucinous neoplasm of the pancreas. Pancreatology. 2023 Dec 28:S1424-3903(23)01883-5.

All the text about this citation should be revised accordingly.

RE: Dear Reviewer, thank you for this valuable suggestion. We have read with great interest the work by Ohtsuka et al., which aimed to revise the 2017 international consensus guidelines for the management of IPMN of the pancreas. We have now added this reference to the text and also introduced some comments regarding this last update of the guidelines. Thank you again for your help in further improve our paper.

3) The authors previously published on the use of pineapple juice for MRI instead of gadolinium. I wonder whether it could be of interest in this setting too.

RE: Dear Reviewer, thank you very much for this insightful observation. A recognised limitation of MRCP is the possible overlap of the static fluids in the pancreatobiliary system and the static fluid in the gastrointestinal tract (i.e. the stomach, duodenum and proximal jejunum) which can obscure the distal portion of the common bile duct or simulate a disease [MR cholangiopancreatography and MR urography: Improved enhancement with a negative oral contrast agent. Radiology. 1997;203(1):281–285. doi: 10.1148/radiology.203.1.9122408]. Several oral negative contrast agents, containing paramagnetic substances which shorten the T2 relaxation time, thus reducing the signal hyperintensity of gastro-enteric fluids in standard MRCP sequences, have been produced by pharmaceutical companies and have been utilized in daily radiological practice in the past in order to overcome this limitation. However, these negative oral contrast agents have many limitations; they are relatively unpalatable, are too diluted in the gastrointestinal tract or are too expensive. To overcome these limitations, in more recent years, natural fruit juices have gained attention, the most popular being pineapple juice (PJ) and blueberry juice, owing to their properties which are similar to pharmaceutical oral negative contrast agents, having the same ability to suppress the high signal from gastrointestinal tract liquids on MRCP, but without the described limitations, having good palatability and a lower cost Pineapple juice as a negative oral contrast agent in magnetic resonance cholangiopancreatography: A preliminary evaluation. Br. J. Radiol. 2004;77(924):991–999. doi: 10.1259/bjr/36674326]. In particular, our previous study has the merit of having demonstrated that the oral administration of 150 ml (one glass) of pineapple juice containing a high concentration of Mn2+ prior to MRCP is sufficient to guarantee the correct Mn2+ concentration to suppress the gastroduodenal liquid signal, without affecting the image quality of the examination [Optimization of pineapple juice amount used as a negative oral contrast agent in magnetic resonance cholangiopancreatography. Sci Rep. 2022 Jan 11;12(1):531. doi: 10.1038/s41598-021-04609-6]. These considerations have now been added to the text.

Round 2

Reviewer 1 Report

Comments and Suggestions for Authors

No Further comments to authors 

Reviewer 2 Report

Comments and Suggestions for Authors

The text has been revised satisfactorily